# Enhanced Antimicrobial Activity of Biocompatible Bacterial Cellulose Films via Dual Synergistic Action of Curcumin and Triangular Silver Nanoplates

**DOI:** 10.3390/ijms232012198

**Published:** 2022-10-13

**Authors:** Eduardo Lanzagorta Garcia, Marija Mojicevic, Dusan Milivojevic, Ivana Aleksic, Sandra Vojnovic, Milena Stevanovic, James Murray, Olivia Adly Attallah, Declan Devine, Margaret Brennan Fournet

**Affiliations:** 1Materials Research Institute, Technological University of the Shannon: Midlands Midwest, N37 HD68 Athlone, Ireland; 2Institute of Molecular Genetics and Genetic Engineering, University of Belgrade, Vojvode Stepe 444a, 11000 Belgrade, Serbia

**Keywords:** bacterial cellulose, silver nanoparticles, curcumin, antimicrobial properties, biopolymers, zebrafish

## Abstract

Curcumin and triangular silver nanoplates (TSNP)-incorporated bacterial cellulose (BC) films present an ideal antimicrobial material for biomedical applications as they afford a complete set of requirements, including a broad range of long-lasting potency and superior efficacy antimicrobial activity, combined with low toxicity. Here, BC was produced by *Komagataeibacter medellinensis* ID13488 strain in the presence of curcumin in the production medium (2 and 10%). TSNP were incorporated in the produced BC/curcumin films using ex situ method (21.34 ppm) and the antimicrobial activity was evaluated against *Escherichia coli* ATCC95922 and *Staphylococcus aureus* ATCC25923 bacterial strains. Biological activity of these natural products was assessed in cytotoxicity assay against lung fibroblasts and in vivo using *Caenorhabditis elegans* and *Danio rerio* as model organisms. Derived films have shown excellent antimicrobial performance with growth inhibition up to 67% for *E. coli* and 95% for *S. aureus*. In a highly positive synergistic interaction, BC films with 10% curcumin and incorporated TSNP have shown reduced toxicity with 80% MRC5 cells survival rate. It was shown that only 100% concentrations of film extracts induce low toxicity effect on model organisms’ development. The combined and synergistic advanced anti-infective functionalities of the curcumin and TSNP incorporated in BC have a high potential for development for application within the clinical setting.

## 1. Introduction

Bacterial cellulose (BC) has gained great attraction from several research and industrial sectors due to its high purity, absence of lignin and hemicellulose commonly found in plant cellulose, and its exclusive properties [1]. The crystalline structure of BC is naturally composed of randomly assembled nanofibrils aggregated in bundles. Such arrangement allows a great surface area that provides BC with high liquid holding capacity, flexibility, mouldability, and high mechanical strength in the wet state [1,2,3]. Furthermore, BC is considered a biocompatible material which renders it suitable for a wide range of applications, especially in the biomedical field, such as wound healing systems [4], BC-based biosensors [5], tissue regeneration [6], scaffolds [7], and transdermal applications [2,3]. However, BC still shows high challenges regarding interactions and functionalization for constructing uniform functional materials. Pristine BC lacks the adhesive sites necessary for migration and cell signaling which can be improved by further modifications such as crosslinking with metal ions [8].

The production process of BC is regarded as one of the crucial steps that controls the properties of the resulting material. Various species of bacteria, including Gram-negative and Gram-positive, and different substrates can be used for BC production. Such variety results in different morphologies, structures, properties, and applications of the produced BC [9]. Process of BC production involves several biochemical reactions and it was revealed that specific operons are in control of biosynthesis, transportation through cells and supramolecular assembly of cellulose fibrils. However, molecular mechanism, more specifically: transport of acyl groups from the Krebs’s cycle to the periplasm, transport of glucan chains from the cell and polymerization of glucose are yet to be determined [10].

Between Gram-positive bacteria only few species have been reported to produce BC, such as: *Bacillus* sp. [11], *Leifsonia* sp. [12], *Lactobacillus* sp. [13], and *Rhodococcus* sp. [14]. In contrast, the number of Gram-negative BC producers is significantly higher. Species belonging to *Pseudomonas* [15], *Escherichia* [16], *Alcaligenes* [17], *Enterobacter* [18], *Salmonella* [19], *Acetobacter* [20], and *Komagataeibacter* genera have been reported as BC producers in the literature. On the other hand, bio-cellulose can be synthesized in a cell-free system in the presence of appropriate enzymatic machinery after disrupting bacterial cell walls [21].

Recently, one of the most attractive genera that has been studied for BC production is *Komagataeibacter*. Species belonging to this genera are well-recognized, exceptionally efficient BC producers known for its phenotype diversity manifested by preference of the carbon source, BC structure and production rate depending on the strain. Several species from this genus have been identified as strong cellulose producers, including *Komagataeibacter xylinus*, *Komagataeibacter medellinensis*, *Komagataeibacter oboediens*, *Komagataeibacter pomaceti*, *Komagataeibacter nataicola*, *Komagataeibacter rhaeticus* [1].

The utilization of other compounds as substrates for BC production, together with glucose has been recently evaluated as an approach for improving the properties of resulting BC, as a final product [22]. Such an approach can also be extended to enhance the antimicrobial activity of BC, given its wide range of applications in the biomedical field. Improvement of polymers’ antimicrobial properties is a popular research topic [23]. Different additives have been proposed to produce BC with enhanced antimicrobial properties. These include the addition of antibiotics [24], inorganic antimicrobials such as metallic nanoparticles [25], carbon nanomaterials [26] and organic antimicrobials such as bioactive substances [27] or synthetic compounds [28]. Moreover, based on reported literature, it was proved that associating two antimicrobial agents with a single composite can have great benefits due to the synergistic action of both compounds to overcome microbial resistance [29,30].

Curcumin is a polyphenolic pigment obtained from the turmeric root plant. It displays a broad-spectrum of antibacterial, antifungal, and antiviral properties which rendering it an attractive candidate for enhancing BC antimicrobial properties [31,32]. Fabrication of curcumin-loaded BC hydrogels has been explored as wound dressing material [33,34,35,36,37] with favorable results of biocompatibility and successful growth inhibition of Gram-negative and Gram-positive strains. Other studied applications include the treatment of melanoma in skin cancer cells [38] and active packaging materials [39]. Studies by Adamczak et al. showed that curcumin clearly has stronger effect against Gram-positive in comparison to Gram-negative bacteria. Curcumin has very selective antimicrobial activity, its effect strongly depends on the microorganism itself, not only genus related, but on the strain level as well [40].

Silver nanoparticles (Ag NPs) are another widely used broad-spectrum antimicrobials that have been incorporated into BC to boost their antimicrobial properties. Different approaches have been undertaken for the fabrication of BC-Ag NPs composites, focusing mainly on in situ methods for the synthesis of the nanoparticles within the BC [41,42,43,44,45], as well as immersion of BC in Ag NPs solution for impregnation of the nanoparticles into the BC matrix [46,47,48]. Wang et al. used chemical oxidation method to prepare silver oxide powder and combined it with cellulose nanofibrils to improve antimicrobial properties of this material [49]. The variation of incorporated Ag NPs morphologies also showed great potential in providing additional or differential properties for BC which can further broaden its applications. For instance, silver nanowires have been incorporated into BC to boost its antimicrobial effects thanks to the higher length-to-diameter ratio of the nanowires, which also prevents blocking the BC pores and allows air circulation for wound dressing applications [50]. Despite such potential of the different shapes of Ag NPs, the fabrication of BC composites using other Ag NP shapes, such as TSNP, which have exhibited higher antimicrobial activity compared to spherical NPs [51,52,53,54], has not been yet explored.

Considering previously stated, and the rising interest of the scientific community and manufacturing sectors, it is clear that there is still room for BC properties improvements. One of the most promising approaches includes designing BC-based composites with superior characteristics, important from the perspective of the desired application field. This study aimed to assess the biocompatibility and antibacterial characteristics of the BC materials produced in the presence of curcumin with ex situ incorporated TSNP.

## 2. Results

### 2.1. Production of BC

*Komagataeibacter medellinensis* ID13488 bacterial strain was used to produce BC in HS medium, and HS medium that was supplemented with 2% and 10% of curcumin (*w*/*v*) (Figure 1). The BC films exhibited clear color differences due to curcumin supplementation using visual inspection. The intensity of the acquired orange color is notably greater for the sample supplemented with 10% curcumin (BC-Cur10%), than the one with 2% (BC-Cur2%).

After 14 days of incubation, the produced BC was dried and weighed. Due to the presence of curcumin, BC pellicles resulted in higher mass compared to the non-supplemented medium (Figure 2). Differences in weights are also a result of curcumin incorporation. UV-Visible spectral measurements showed that BC films absorbed 27.8% and 41% of curcumin during the two-week incubation period. 

### 2.2. Characterization of Produced BC Films

#### 2.2.1. Scanning Electron Microscopy (SEM) and Energy Dispersive X-ray Spectroscopy (EDS)

SEM micrographs of the dried BC films grown in the presence of curcumin are shown in Figure 3. A smooth surface was observed in bare BC films (Figure 3A), while samples with incorporated curcumin showed crystal particles on the materials’ surface (Figure 3C,E). The presence of curcumin crystals indicates their strong interaction with the nanofibril matrix of BC [55].

The distribution of TSNP appearing as bright white dots on the surface of BC films is illustrated in Figure 4. The percentage of TSNP within the BC films was determined with the help of EDS analysis, 5 different white dots were randomly selected and analyzed. The presence of silver (Ag) was confirmed for all the selected spots in tested samples (BC-TSNP-Cur2%, BC-TSNP-Cur10%). EDS analysis revealed a strong signal in the silver region with the average percentages of 80.56 and 63.9% for examined materials, respectively.

#### 2.2.2. Thermal Gravimetric Analysis (TGA)

The TGA analysis was carried out to study the thermal stability of the fabricated BC films. The TGA curves of bare BC and BC-Cur and BC-Cur-TSNP films are shown in Figure 5. All the films started gradually losing mass below 100 °C, which could be attributed to the moisture content in the BC films. Maximum mass loss occurred between 200 °C and 600 °C. For most samples, the onset degradation temperature (T_onset_) was around 213 °C, and the degradation rate was stabilized at 600 °C. Furthermore, it can be indicated that the incorporation of curcumin and TSNP did not significantly affect the BC films’ thermal stability, and this outcome follows the reported literature [25,56].

#### 2.2.3. Fourier Transform Infrared Spectroscopy (FTIR)

FTIR analysis was performed to obtain a comparative view of the functional groups in BC and their changes after the addition of curcumin and TSNP. The resulting FTIR spectra for the BC specimens are shown in Figure 6. The spectrum of bare BC showed characteristic peaks at 3277 cm^−1^ (O-H stretching vibration), 2918 cm^−1^ (C-H stretching), 1626 cm^−1^, 1535 cm^−1^ (protein amide II absorption), 1156 cm^−1^ (C-O-C asymmetric bending/stretching), 1055–1033 cm^−1^ (bending of C-O-H bond of carbohydrates), and 665 cm^−1^ (C-OH out of plane bending) [57,58]. The spectrum for curcumin allowed to identify characteristic peaks at 3504 cm^−1^ (O-H stretching vibration), 1600 cm^−1^ (benzene ring stretching vibration), 1503 cm^−1^ (C=O and C=C vibrations, 1426 cm^−1^ (olefinic C-H bending vibrations), 1272 cm^−1^ (aromatic C-O stretching vibrations), and 1026 cm^−1^ [59,60,61,62].

Spectrum of BC-Cur10% shows characteristic peaks from both, BC and curcumin. As shown in Figure 6, the peak assigned to OH stretching vibration of BC exhibited a shift from 3340 cm^−1^ to 3346 cm^−1^, and the peak assigned to C=O and C=C vibrations of curcumin, shifted from 1503 cm^−1^ to 1507 cm^−1^. A weak peak at 1279 cm^−1^ was attributed to C-H bending in the spectrum of BC [63], while it was much stronger in the same wavelength for BC-Cur10%. The small peaks that showed from 1550 cm^−1^ to 1100 cm^−1^ in the BC spectrum were masked by larger curcumin peaks in the spectrum of BC-Cur10%. The rest of the peaks either remained in the same wavelength or had a small shift of 1–2 cm^−1^ only, when compared to BC or curcumin spectra. Thus, the obtained results confirm the incorporation of curcumin into the BC matrix and suggest a chemical interaction with the cellulose fibrils. A similar tendency was observed by Sajjad et al. after incorporating curcumin into BC sheets, where additional peaks appeared, and the characteristic peaks of BC suffered small shifts [36].

In the spectrum obtained for BC-TSNP-Cur10%, a shift from 3340 to 3277 cm^−1^ was observed in the OH peak. The C=O and C=C vibrations peak was also shifted from 1503 to 1512 cm^−1^ concerning the curcumin spectrum. Additionally, when compared to the spectrum of BC-Cur10%, several peaks showed reduced intensities between 1500 and 1000 cm^−1^, which are estimated to occur due to the impregnation of TSNP within the matrix of BC-Cur films.

#### 2.2.4. Weathering Testing

When combined with humidity, the color of an organic pigment can fade much more quickly than due to light alone. For this reason, weather testing is often carried out to measure irradiation effects with humidity. Curcumin is a very light-fast material and its color fades quickly when exposed to such conditions. Even with the addition of UV absorbers and hindered amine light stabilizers, curcumin remains relatively unstable.

Figure 7 shows warping occurred in all five samples after weather testing. This indicates that the combination of UV and moisture caused structural changes to the BC film, presumably due to reduced chain length from photooxidative degradation on the exposed surface, which was more than that on the rear of the samples. The weathering cycles caused embrittlement of samples containing curcumin, without TSNP. Large deposits of curcumin were present on the surface of the samples, indicating that perhaps much of the curcumin is rejected from the wafer as the solution dries. The degree of incompatibility between the BC and curcumin was less pronounced in the samples containing TSNP. While it is not clear from Figure 7, spectrophotometry measurements showed color fading in all samples. The values for color changes due to weathering exposure are presented in Table 1 as ΔE* to calculate color difference. Due to the initial color differences between samples, it is not fair to directly compare ΔE* values between cases; however, it is clear that none of the materials is UV stable. Not only is there a significant color change for each case, but the embrittlement and disintegration of the material indicates a likely deterioration in mechanical properties due to UV light exposure. Such behavior will be addressed and resolved in future work.

### 2.3. Biological Evaluations

#### 2.3.1. Antimicrobial Activity

Antimicrobial activity of derived BC films was evaluated against *S. aureus* and *E. coli*, as representatives of Gram-positive and Gram-negative bacteria, respectively. Results are displayed in Figure 8 as bacterial growth percentage. Pure BC film did not affect the growth of the tested strains in the desired fashion; on the contrary, the growth was enhanced by approximately 10% in both bacterial strains. Nevertheless, the antimicrobial performance of the BC films was significantly enhanced in the presence of curcumin, reducing the growth by approximately 15% in BC-Cur2% film, and up to 33% with BC-Cur10% films. Additional incorporation of TSNP in BC-Cur films resulted in further inhibition of *E. coli* growth (up to 67% of reduction) and even greater inhibition against S. aureus (up to 95% of growth reduction). The statistical analysis by ANOVA-Dunnett (α = 0.05) showed that for *E. coli*, all the groups are significantly different to the control, with the exception of BC and BC-Cur2%. Meanwhile, in the case of *S. aureus*, the only group that did not result to be significantly different to the control, was the bare BC.

#### 2.3.2. Cytotoxicity Assay

Cytotoxicity was evaluated by using a human fibroblasts cell line (MRC5) exposed to BC film extracts of different concentrations, previously prepared in RPMI medium under dynamic conditions. The viability of MRC5 cells was assessed using a standard MTT assay. The results presented in Figure 9 indicate that more than 80% of cells survived after being in contact with 100% concentrations of BC film extracts, indicating their low toxicity. According to the literature, materials are considered safe when the cells viability is over 70% [64]. It was also shown that only 100% concentrations of BC-Cur2% and BC-Cur10% film extracts induce a significant decrease in MRC5 cells viability after 48 h treatments. Significant toxicity was related to 100% and 50% concentrations of both BC-TSNP and BC-TSNP-Cur2%. Interestingly, the increased amounts of curcumin in BC-TSNP-Cur10% film extract resulted in considerably reduced toxicity since more than 80% of cells survived after being in contact with the highest concentration of this extract. The statistical analysis by ANOVA-Tukey (α = 0.05) showed that for the extracts in a concentration of 100%, showed a significant difference of all the materials against the bare BC survival growth, except for the BC-TSNP-Cur10% group. Meanwhile, in the 50% extracts, the groups of BC-TSNP and BC-TSNP-Cur2% were the only ones that showed a statistical significance.

#### 2.3.3. *Danio Rerio* Embryotoxicity Assay

The effects of BC materials were examined on zebrafish embryos in 12.5%, 10%, 5%, and 2.5% concentrations (Figure 10).

As presented in Figure 11, control material BC was not significantly toxic, while BC TSNP in the highest tested concentration (50%) was toxic, with the teratogenic effects such as microcephaly and hepatotoxicity. Supplementation of BC with curcumin in two concentrations causes toxic effects on zebrafish, mostly heart and liver toxicity, while BC-TSNP-Cur2% and BC-TSNP-Cur10% did not have any toxic or teratogen effects on zebrafish embryos.

#### 2.3.4. *Caenorhabditis elegans* Survival Assay

In order to confirm previous results, the effects of BC materials were examined on *C. elegans* model system in four different concentrations: 50, 25, 12.5, and 6.25 µg mL^−1^ and the results are presented in Figure 12. *C. elegans* has been widely used to evaluate biological activity and interactions of nanoparticles and natural products with different targets in organisms [65,66]. Overall BC materials showed little to no toxicity, with only around 10% dead at the highest concentration for BC-TSNP-Cur2%, BC-TSNP, and BC-Cur10%.

## 3. Discussion

Advanced anti-infective biomedical materials which meet the stringent requirement for applications in clinical settings are urgently required [67]. BC is a sustainable biopolymer with unique physical properties leading to its expansive potential in the biomedical field. Effectively harnessing and integrating the functional properties of natural materials for synergistic performance, development has the potential to deliver high-performance biomaterials. Here a series of high-impact results have been delivered by developing enhanced antimicrobial curcumin and TSNP-incorporated BC films.

First, the production of BC in the presence of curcumin resulted in a higher product yield, up to 200% in the case where 10% of curcumin was used in the medium. This result is highly significant considering the current multiple inhibiting factors within BC production and consumption. Moreover, the addition of curcumin has improved the antimicrobial characteristics of the resulting material. Such results could be attributed to the antimicrobial activity of curcumin itself, which boosted the overall antimicrobial activity of the BC-Cur films. The antimicrobial activity was also concentration dependent since it increased with the increase in curcumin amount added during the cultivation of BC. These observations coincide with previously published results by Sajjad et al. (2020) [36]. This antimicrobial effect was enhanced further again with the incorporation of TSNP, resulting in inhibition of *E. coli* growth up to 67% and even greater inhibition against S. aureus (up to 95% of growth reduction). Contrary to our results, Gupta et al. (2020) observed higher growth inhibition against the Gram-negative strain than the Gram-positive, when testing the antimicrobial activity for ex situ curcumin-loaded BC [68]. Nonetheless, curcumin has previously been demonstrated to exhibit a higher antimicrobial effect toward Gram-positive strains than Gram-negative. Minimal inhibitory concentration of crude curcumin against *S. aureus* was found to be ~100 µg mL^−1^, while for Gram-negative *E. coli* this concentration increases four-fold. This suggests that the activity of curcumin is predominant in the case of Cur-TSNP-loaded BC films against Gram-positive strains [32,40,69].

In a highly positive synergistic interaction, BC films with 10% curcumin and incorporated TSNP have shown reduced toxicity with an 80% cells survival rate. Achievement of this feature is important because the material can be characterized as safe once cells viability is over 70%, and it can be concluded that modified BC is a biocompatible material [64]. In vivo tests further confirmed these results. In the case of the zebrafish model system, supplementation of BC with curcumin in both used concentrations caused heart and liver toxicity, while BC films with curcumin and TSNP did not have any toxic or teratogen effects on zebrafish embryos. Previously published study, where curcumin was tested separately on zebrafish embryos, showed the severe toxicity of this compound, with some of the teratogenic effects including bent or hook-like tails, spinal column curving, edema in pericardial sac, retarded yolk sac resorption, and shorter body length in the examined concentration of 7.5 µmol L^−1^ [70]. While in a different study, tested in lower dosages (1–6 µmol L^−1^), curcumin had a moderate acute toxicity effect on zebrafishes with no obvious morphological abnormalities according to their score based on the degree of morphological anomalies (2–4 minor toxic effects) [71]. In addition, silver nanoparticles are toxic for zebrafish development; one study revealed that nanoparticle treatments resulted in concentration-dependent toxicity, typified by phenotypes that had abnormal body axes, twisted notochord, slow blood flow, pericardial edema, and cardiac arrhythmia [72]. Our study suggests that combining curcumin and TSNP in BC does not show high toxicity on zebrafish development in the examined conditions. The synergetic effect of these natural compounds leading to improved biological activity of the BC films obtained in this study will be further addressed in the future. In the case of *C. elegans*, previous studies on curcumin itself reported that this compound can prolong lifespan and influence age-related biomarkers in this model organism [73]. On the other hand, nanoparticles, especially silver, have shown significant toxic effects on this model organism [74]. Our results imply that the concentration of TSNP incorporated in BC films was insufficient to affect *C. elegans* under examined conditions significantly.

BC is a biomaterial of growing importance with a rising application spectrum and its developments are very significant. Further studies will include improvement of mechanical characteristics of these materials that will potentially result in designing products suitable for various packaging and biomedical applications. This study has effectively demonstrated that integrating curcumin in a production medium with ex situ TSNP incorporation leads to safe, biocompatible, antimicrobial BC films.

## 4. Materials and Methods

### 4.1. Materials

The HPLC-grade water (34877-2.5L), sodium citrate tribasic (C8532-100G) [TSC], poly(sodium 4-styrenesulfate) (434574-5G) [PSSS], sodium borohydride (213462-25G) [NaBH_4_], silver nitrate (204390-10G) [AgNO_3_], L-ascorbic acid (A92902-25G) [AA] and citric acid (C2404-500G) were obtained from Sigma-Aldrich Ireland Ltd. (Arklow, Ireland). LP-BT100-1F Peristaltic Dispensing Pump with YZII15 pump head and Tygon LMT-55 Tubing #17 were obtained from Drifton (Hvidovre, Denmark). Curcumin (FC09321) was obtained from Biosynth Carbosynth (Thal, Switzerland). Glucose (NCM0241A), yeast extract (NCM0218A), and balanced peptone No. 1 (NCM0257A) were manufactured by Neogen (Lansing, MI, USA) and distributed by Cruinn Diagnostics (Dublin, Ireland). Sodium phosphate dibasic dodecahydrate (71650) [Na_2_HPO_4_] was obtained from Fluka Analytical.

### 4.2. Synthesis of TSNPs

TSNP were synthesized by a previously described seed-mediated approach [75]. Briefly, NaNH_4_ (0.57 mL, 10 mM) was used for the reduction of AgNO_3_ (9.48 mL, 0.5 mM), using TSC (9.5 mL, 2.5 mM) and PSSS (0.47 mL, 500 mg/L) as stabilizers for the formation of the seeds. TSNP were afterwards grown from the seeds (8.75 mL, 25.56 ppm), with the addition of 100 mL of distilled water, AgNO_3_ (75 mL, 0.5 mM) and using AA (1.87 mL, 10 mM) as a reducing agent. TSC (12.5 mL, 25 mM) was added at the end of the reaction to provide stabilization to the TSNP.

### 4.3. Production of BC Films Using Curcumin as a Supplement

The BC was produced using *Komagataeibacter medellinensis* ID13488. Pre-culture of the bacteria was prepared in Hestrin and Schramm (HS) liquid medium (2% glucose, 0.5% yeast extract, 0.27% Na_2_HPO_4_ and 0.15% citric acid) as previously reported [76], and the incubation was carried out for 2 days at 30 °C on a rotary shaker with the agitation rate of 180 rpm. BC production was performed in HS medium supplemented with 2% and 10% curcumin (*w*/*v*). BC films grown in HS medium without additives were used as a negative control for further assessments.

Different media batches were inoculated with 100 μL mL^−1^ from pre-culture (10% *v*/*v* inoculum), and incubated statically at 30 °C for 14 days. After incubation, BC hydrogels were removed from liquid media. Bare BC films were washed in a 0.5 M potassium hydroxide (KOH) solution in the water bath (1 h, 100 °C) and neutralized with distilled water. In order to prevent degradation of curcumin, BC films produced with the addition with curcumin were immersed in distilled water and autoclaved (121 °C, 15 min). After washing process, films were dried overnight at 60 °C.

#### Estimation of Curcumin Absorption from Media

The UV-visible spectral measurements were performed in a UV 1280 Shimadzu (Kyoto, Japan) spectrophotometer using 10 mm path length matched quartz cuvettes [77]. The measurements were taken in the wavelength range of 200–700 nm. In aqueous solution, curcumin had a peak at 427 and a shoulder at 360 nm. Peak area was used for estimation of curcumin incorporation via comparison between non-inoculated media and media after 14 days of incubation.

### 4.4. Preparation of TSNP Incorporated BC Films

A total of 20 mg of each examined BC films (BC with 2% of curcumin, BC with 10% of curcumin) were immersed into 5 mL of TSNP solution (21.34 ppm) and incubated overnight at 30 °C with shaking on horizontal platform (100 rpm). Afterwards, films were removed from the solution and dried overnight at 60 °C.

### 4.5. Characterisation of Derived BC Films

#### 4.5.1. Morphological Characterization and Estimation of TSNP Absorption

Scanning electron microscopy (SEM) images were obtained using Mira XMU SEM (Tescan™, Brno, Czech Republic) in back scattered electron mode for surface analysis. The accelerating voltage used was 9 kV. Prior to analysis, tested samples were placed on an aluminum stub and sputtered with a thin layer of gold using Baltec SCD 005 for 110 s at 0.1 mbar vacuum. Energy dispersive X-ray spectroscopy (EDS) was used to confirm the presence of silver in the samples immersed in TSNP solution. Data were gathered by an X-Max E.D.S. system (Oxford Instruments, Oxford, UK) with Inca software.

#### 4.5.2. Thermogravimetric Analysis (TGA)

Thermal stability of the BC films was evaluated using a Pyris TGA 1 thermogravimetric analyzer (Perkin Elmer, Washington, DC, USA) with software Pyris 1. The film samples were taken in a standard aluminum pan and heated from 30 to 700 °C at the rate of 10 °C min^−1^ under a nitrogen flow of 50 mL min^−1^.

#### 4.5.3. Fourier Transforms Infrared (FTIR) Spectroscopy

A Perkin-Elmer Spectrum One FTIR spectrometer (Perkin Elmer Inc., Washington, DC, USA) fitted with a universal ATR sampling accessory and Perkin Elmer software, was used to record the spectra of dried BC films. The spectral resolution was 4 cm^−1^ and 20 scans were acquired for each spectrum (4000–650 cm^−1^).

#### 4.5.4. Evaluation of BC Films Stability via Weathering Tests

Weather testing was carried out on the BC films to determine their stability against UV irradiation using a QUV/se accelerated weather tester from Q-Lab (Westlake, OH, USA) in compliance with the ISO 4892 test standard. Fluorescent UV-A lamps with a radiant emission range of 365 nm down to 295 nm were used to simulate sunlight exposure. A set temperature of 40 °C was used in the chamber and UV irradiance of 0.76 W m^−2^. Samples were exposed to moisture spray and UV light cyclically in 4-h increments. In total, the samples were exposed for 20 h in each condition.

To quantify the effects of weathering, spectrophotometry was used before and after exposure to measure the degree of color change in samples. A CM-3610A bench-top spectrophotometer from Konica Minolta (Tokyo, Japan) with an 8° observant angle was used to measure the color change in specimens. Procedures followed the ISO 11664-4 test standard and the ΔE* (CIEDE2000) metric was used to calculate color difference.

#### 4.5.5. Biological Evaluations

##### Antibacterial Activity of BC Films

All the dried BC films’ antimicrobial activity was evaluated against *Escherichia coli* ATCC 95922 and *Staphylococcus aureus* ATCC 25923 in Luria-Bertani (LB) broth (10 g L^−1^ tryptone, 10 g L^−1^ NaCl, 5 g L^−1^ yeast extract, pH 7.2). Overnight cultures of the bacteria were diluted to a concentration of 10^8^ CFU mL^−1^ to be used as pre-culture. Dried BC films (10 mg of each specimen) were immersed in fresh LB broth and inoculated with 1% (*v*/*v*) from pre-culture, for a final concentration of 10^6^ CFU mL^−1^ of bacteria. Untreated BC films were used as negative control. After incubation for 24 h at 37 °C and 100 rpm, BC films were removed from the cultures and optical density (OD) of the cultivated broth was measured at 630 nm using a Biotek Synergy HT Microplate Reader (Biotek Instruments GmbH, Bad Friedrichshall, Germany). Growth percentage was calculated using the following equation (Equation (1)).
(1)$$\mathrm{Growth Percentage}=\left(\frac{\mathrm{Absorbance}\;\mathrm{of}\;\mathrm{tested}\;\mathrm{sample}}{\mathrm{Absorbance}\;\mathrm{of}\;\mathrm{positive}\;\mathrm{control}}\right)\times 100$$

##### Cytotoxicity Assay

The cytotoxicity of BC films was evaluated by testing against the human fibroblasts (MRC-5) obtained from ATCC. BC films were sterilized under UV light for 20 min and then immersed into RPMI (Roswell Park Memorial Institute) medium (5 mg mL^−1^). The samples were incubated at 37 °C, 180 rpm for 48 h. After incubation, samples were centrifuged (5000 rpm, 15 min), and sterilized by filtering using 0.22 µm pore size filters.

Cells were plated into a flat-bottom 96-well plate at a concentration of 1 × 104 cells per well in RPMI medium supplemented with 100 µg mL^−1^ streptomycin, 100 U mL^−1^ penicillin, and 10% (*v*/*v*) fetal bovine serum (FBS) and incubated for 24 h at 37 °C to allow the formation of a monolayer. After 24 h of cells incubation, RPMI medium was substituted with decreasing concentrations of BC extracts: 100%, 50%, 25%, and 12.5% (*v*/*v*) while the control samples contained only RPMI medium. The incubation at 37 °C, 5% CO_2_ continued for 48 h and the cytotoxicity was determined afterwards using 3-(4,5-dimethylthiazol-2-yl)-2,5-diphenyltetrazoliumbromide (MTT) reduction assay [78]. 

The extent of MTT reduction to formazan was measured spectrophotometrically at 540 nm using a Tecan Infinite 200 Pro multiplate reader (Tecan Group Ltd., Mannedorf, Switzerland). The results were presented as percentage of the control (untreated cells) that was arbitrarily set to 100%.

##### *Danio rerio* Embryotoxicity Assay

BC materials where incubated in embryo water (100 mg mL^−1^) 24 h at 30 °C with shaking on horizontal platform (180 rpm), and that extract was used for toxicology assessment. In vivo toxicity evaluation was carried out on the zebrafish (*Danio rerio*) model, and the general rules of the OECD Guidelines for the testing of chemicals were followed while zebrafish embryotoxicity assay was performed [79]. Briefly, zebrafish embryos were produced by mating of adult females and males (ratio 1:2), collected, washed from detritus and distributed into 24-well plates containing 10 embryos per well in 1 mL embryo water (0.2 g L^−1^ of Instant Ocean^®^ Salt in distilled water), and incubated at 28 °C. Experiments were performed in triplicate using 30 embryos per each tested concentration. Non-treated embryos were used as a negative control. Embryos were examined under the stereomicroscope (SMZ143-N2GG, Motic, Germany) every 24 h for five days, counting and discarding dead embryos and observing teratogenic effects. After the assay embryos were killed by freezing at −20 °C for 24 h. All experiments involving zebrafish were performed in compliance with the European directive 2010/63/EU and the ethical guidelines of the Guide for Care and Use of Laboratory Animals of the Institute of Molecular Genetics and Genetic Engineering, University of Belgrade.

##### *Caenorhabditis elegans* Survival Assay

The nematode *Caenorhabditis elegans* AU37, obtained from the Caenorhabditis Genetics Center (CGC), University of Minnesota, Minneapolis, Minnesota, US. *C. elegans* AU37 (glp-4; sek-1), was used to establish the toxicity of BC materials.

For this assay 25 mg of each BC films was suspended in 2.5 mL of M9 medium (45 mL of 5xM9 salt, 50 mL of glucose solution 50% (*w*/*v*), 500 µL of 1 M MgSO_4_, 25 µL of 1 M CaCl_2_, 250 µL vitamins, 250 µL of trace elements solution, 125 µL of ampicillin 50 µg mL^−1^, and final volume adjusted to 250 mL with double distilled water). Samples were incubated at 37 °C, with shaking on horizontal platform at 180 rpm for 24 h. After incubation, BC films were removed from the suspension and extracts were prepared in the following concentrations: 50 µg mL^−1^, 25 µg mL^−1^, 12.5 µg mL^−1^, and 6.25 µg mL^−1^.

The worm was propagated under standard conditions, synchronized by hypochlorite bleaching, and cultured on a nematode growth medium using *E. coli* OP50 as a food source, as described previously [80]. The *C. elegans* AU37 survival assay followed the standard procedure [81] with some minor modifications. Briefly, synchronized worms (L4 stage) were suspended in a medium containing 95% M9 buffer (3.0 g of KH_2_PO_4_, 6.0 g of Na_2_HPO_4_, 5.0 g of NaCl, and 1 mL of 1 M MgSO_4_∙7H_2_O in 1 L of water), 5% LB broth (Oxoid, Basingstoke, UK), and 10 μg mL^−1^ of cholesterol (Sigma-Aldrich, Munich, Germany). The experiment was carried out in 96-well flat-bottomed microtiter plates (Sarstedt, Nümbrecht, Germany) with a final volume of 100 μL per well. The wells were filled with 50 μL of the suspension of nematodes (25–35 nematodes) and 50 μL of tested BC film suspension. Subsequently, the plates were incubated at 25 °C for 48 h. The fraction of dead worms was determined after 48 h by counting the number of dead worms and the total number of worms in each well, using a stereomicroscope (SMZ143-N2GG, Motic, Germany). All BC films suspensions were tested three times in each assay, and each assay was repeated two times (*n* = 6). As a negative control experiment, nematodes were exposed to the medium containing 1% DMSO.

### 4.6. Statistical Analysys

Statistical analysis for the in vitro cytotoxic evaluation was performed using one-way analysis of variance (ANOVA) and grouped using the Tukey method with a 95% confidence interval. Statistical analysis for the antimicrobial evaluation was performed using ANOVA, paired with the Dunnett method and a 95% confidence interval. The software used to perform these analyses was Minitab (Version 20.2) for Windows (64-bit). All data collected for this study were expressed as mean ± standard deviation. A sample size of four was used for each concentration tested in the cytotoxicity evaluation, and a sample size of three was used for the antimicrobial evaluation.

## Figures and Tables

**Figure 1 ijms-23-12198-f001:**
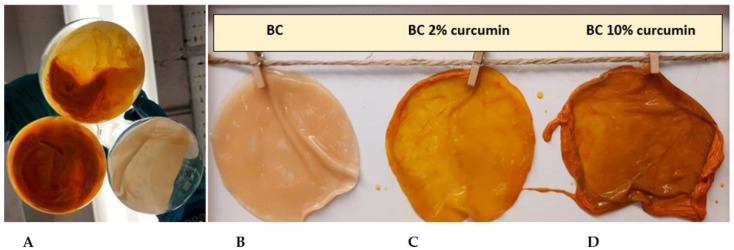
(**A**): Difference between BC pellicles in HS and curcumin-modified HS media after 14 days of incubation; (**B**): BC produced in HS medium; (**C**): BC produced in 2% curcumin supplemented HS medium; (**D**): BC produced in 10% curcumin supplemented HS medium.

**Figure 2 ijms-23-12198-f002:**
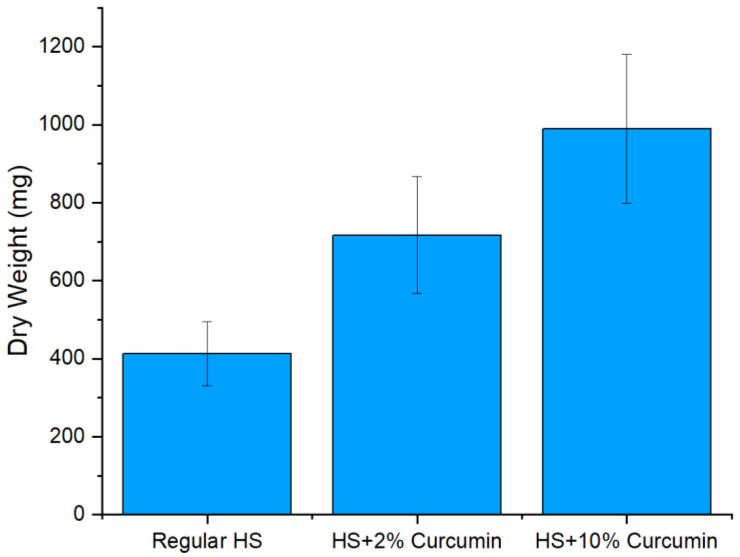
Dry weight of BC and curcumin-supplemented BC with 2% and 10%.

**Figure 3 ijms-23-12198-f003:**
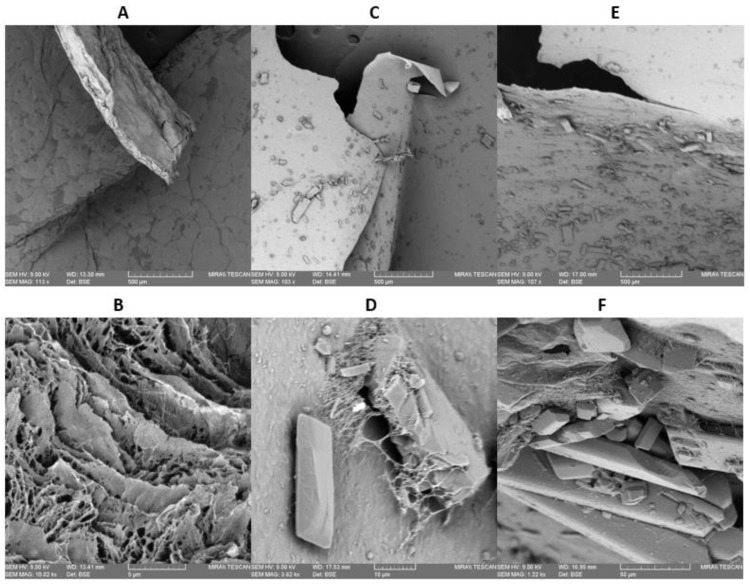
SEM micrographs of BC film 100× (**A**) and 10k× (**B**); BC-Cur2% 100× (**C**) and 3.5k× (**D**); BC-Cur10% 100× (**E**) and 1.2k× (**F**).

**Figure 4 ijms-23-12198-f004:**
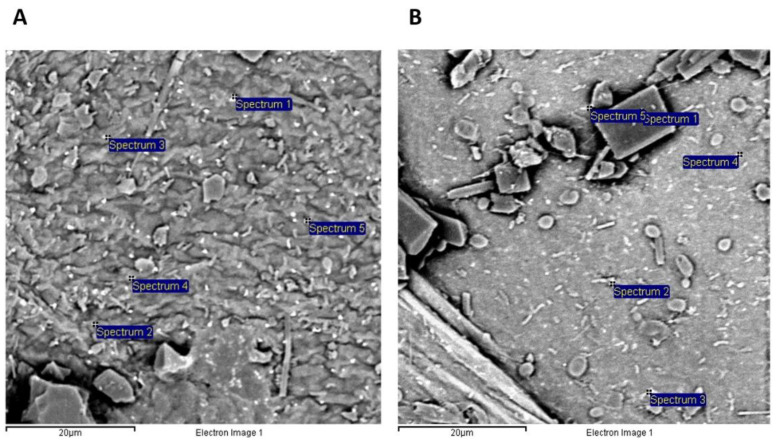
SEM micrographs of examined materials with spots selected for EDS analysis: BC-TSNP-Cur2% (**A**) and BC-TSNP-Cur10% (**B**).

**Figure 5 ijms-23-12198-f005:**
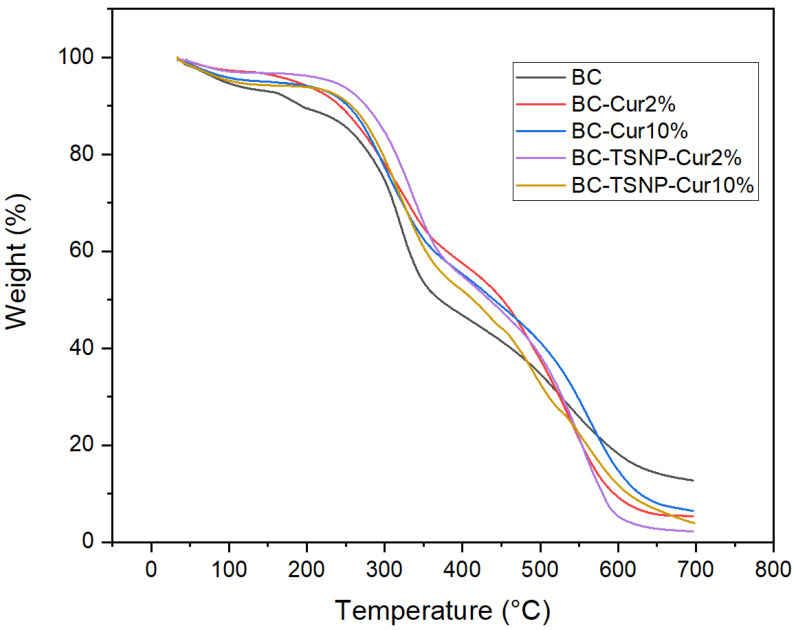
TGA curves of BC, Cur-treated BC, and Cur-TSNP-treated BC films.

**Figure 6 ijms-23-12198-f006:**
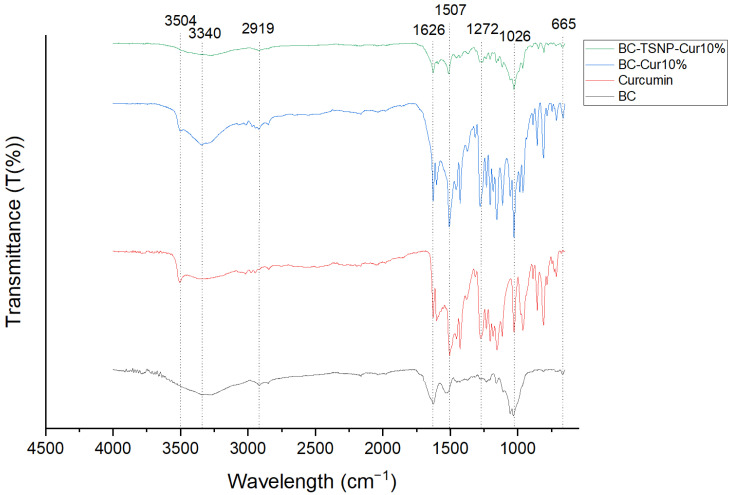
FTIR spectra for BC, curcumin, BC-Cur10%, BC-TSNP, and BC-TSNP-Cur10%.

**Figure 7 ijms-23-12198-f007:**
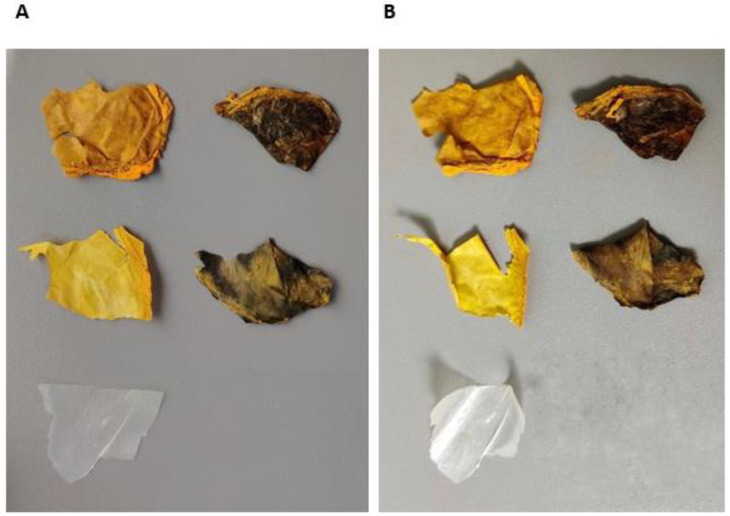
Samples before (**A**) and after (**B**) weather testing. From left-to-right, top-to-bottom: BC-Cur10%, BC-TSNP-Cur10%, BC-Cur2%, BC-TSNP-Cur2% and BC.

**Figure 8 ijms-23-12198-f008:**
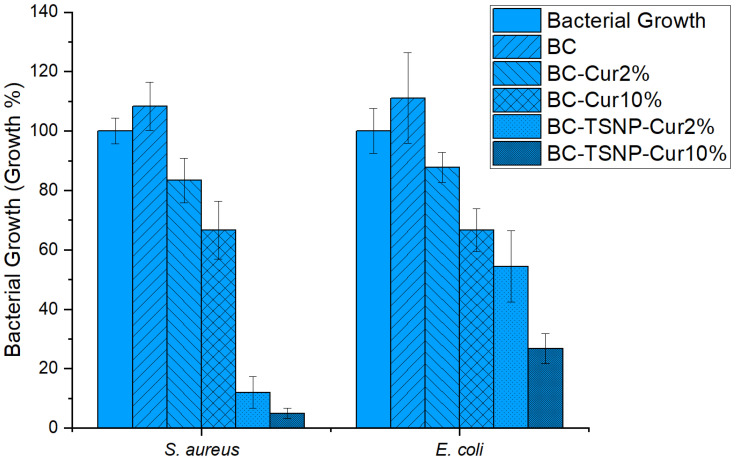
Antimicrobial activity of derived BC films measured as absorbance rate at 630 nm.

**Figure 9 ijms-23-12198-f009:**
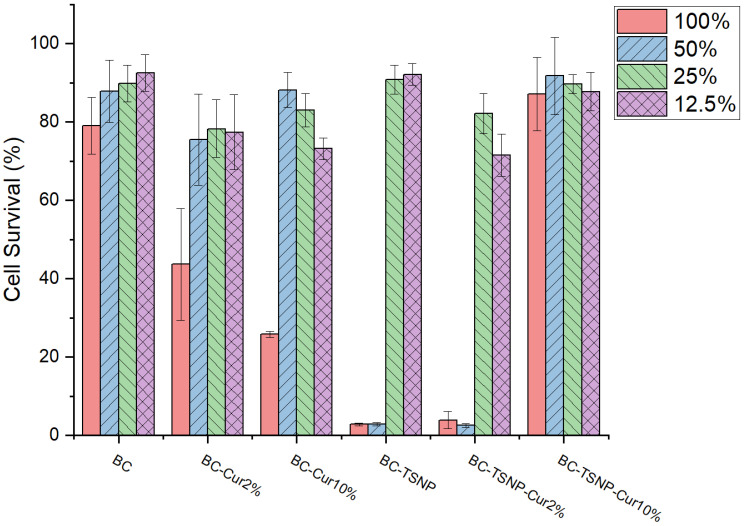
Survival rate of cell line (MRC5) after exposure to BC film extracts of different concentrations.

**Figure 10 ijms-23-12198-f010:**
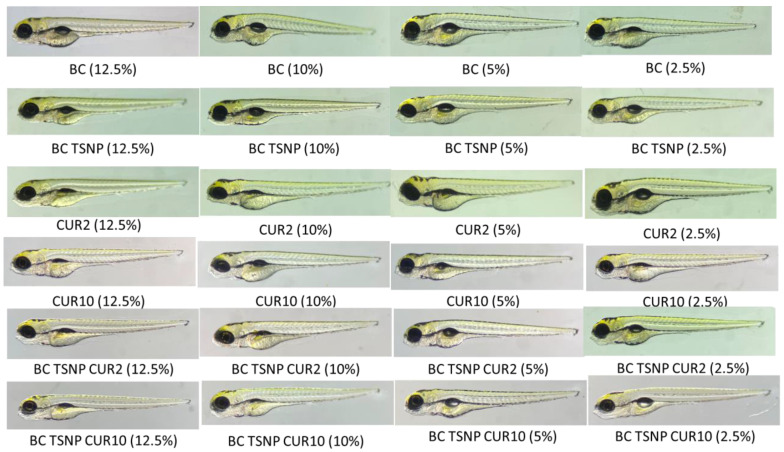
Effect of BC materials on development of zebrafish embryos, images of zebrafish embryos at 120 hpf treated with different extracts concentrations and untreated as a control.

**Figure 11 ijms-23-12198-f011:**
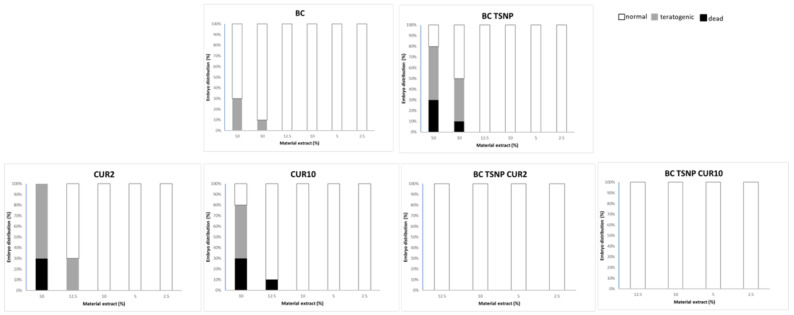
Diagrams for each tested material extract with the percentages of live, dead, and teratogenic embryos.

**Figure 12 ijms-23-12198-f012:**
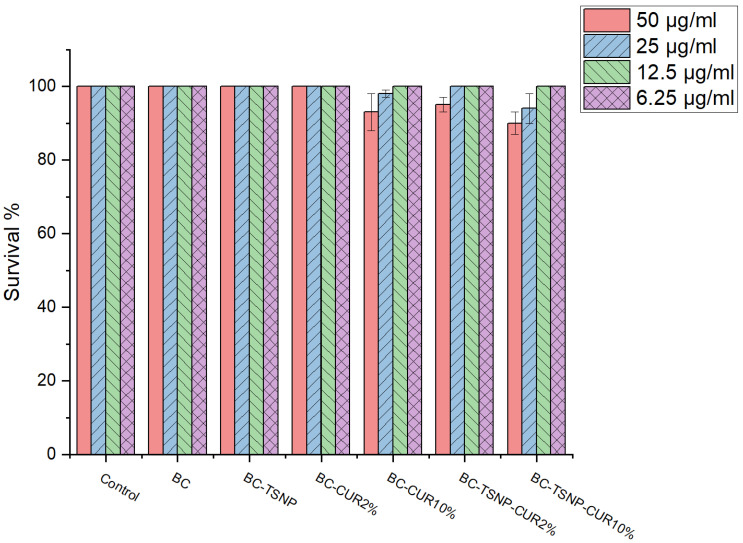
*C. elegans* survival rate in the presence of examined BC materials after 24 h.

**Table 1 ijms-23-12198-t001:** Color change, ΔE* for each sample case due to weather testing.

Samples	ΔE*
BC	2.83
BC-Cur2%	12.74
BC-Cur10%	6.40
BC-TSNP-Cur2%	4.69
BC-TSNP-Cur10%	6.87

## Data Availability

Not applicable.

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
