# Peer review of "Enhanced Antimicrobial Activity of Biocompatible Bacterial Cellulose Films via Dual Synergistic Action of Curcumin and Triangular Silver Nanoplates"

_ijms, 2022, doi:10.3390/ijms232012198_

Round 1

Reviewer 1 Report

In this study, the authors have incorporated two antimicrobial agents, curcumin and Ag NPs, into the BC matrix. They claimed to in situ and ex situ add curcumin and AgNP into the BC matrix, respectively. The developed composites of BC with 2% and 10% curcumin and AgNPs were evaluated for morphological and physico-chemical properties. The antibacterial activity and biocompatibility of the composites were also evaluated. The study is interesting; however, there are some fundamental issues in study design and data presentation, which should be addressed.

1.      At several instances in the manuscript, it is mentioned that curcumin has antibacterial activity. But then authors claimed to in situ develop BC/Curcumin nanocomposite. I wonder how K. medellinensis produce BC in the presence of curcumin in HS medium? Does it mean curcumin has selective antibacterial activity to one strain and no activity against other strain - a clear explanation is required.

2.      Why did you choose 2% and 10% curcumin? The difference in selected concentrations is too high.

3.      At page 3, the authors claimed that curcumin was used as the carbon source that resulted in high BC production. Did you verify this? Apparently, the increased mass of BC produced in the presence of curcumin could be due to the presence of curcumin in the BC matrix that has caused increased BC mass. A clear explanation and interpretation is required.

4.      In the title and elsewhere, authors claimed to develop ‘Triangular-shaped’ AgNP; however, I can see spherical to oval-shaped AgNPs from SEM micrographs. This is a serious issue as this is one of the main aspects of this study claimed by the authors as triangular-shaped AgNP have superior antibacterial activity than spherical nanoparticles.

5.      Abstract is not written well and is too descriptive. It is suggested to add information on the amount/concentration of curcumin and AgNP added to BC as well as quantitative results of the study.

6.      Manuscript is sprinkled with grammatical errors, typos, lacking punctuation, and the sentence structure is odd on several occasions. Unfortunately, due to lack of line numbering, it is hard to point out each mistake. Thorough proofreading is strongly recommended.

7.      Introduction needs improvement according to the below suggestions.

-          1st passage (Lines 3-4) and 2nd passage (Lines 1-2): authors stated that the BC production process and assembly of cellulose fibers determine the crystalline structure and other properties. This is true; however, this description should be refined further by adding a rational discussion about the Molecular regulation of biosynthesis, supramolecular assembly, and tailored structural and functional properties of bacterial cellulose.

-          BC is considered biocompatible due to its non-toxic nature; otherwise, pristine BC lacks adhesive sites to support cell adhesion. Refine this description.

-          1st passage: the biomedical application window of BC should be expanded to biosensing (such as by immobilizing phage as the sensing materials in BC matrix for detection of bacteria). Also provide specific references for each of the mentioned biomedical applications.

-          2nd passage: Provide specific examples of Gram-positive and Gram-negative bacteria producing BC. Also add that in addition to different bacteria, BC (bio-cellulose) is also produced by the cell-free enzymes system and provide references.

-          3rd passage (lines 4-8): ‘not techniques’ but different materials used for making BC-based composites. Correction is required. Also provide specific references for different materials mentioned here.

8.      If allowed by the journal, it is suggested to combine the ‘Results’ and ‘Discussion’ sections.

9.      Figure 1. Label the sub-figures as A, B, C, …. And describe in figure caption.

10.  Figure 5. Y-axis represents the ‘weight (%)’ or’ weight loss (%)’. Please specify.

11.  Why BC production was carried out for 14 days? In general, a 7-10 days incubation is sufficient for BC production as all available carbon sources are utilized by the bacterial cells during this period.

12.  Figure 8. It is suggested to perform the antibacterial activity test through disc diffusion method.  

13.  Other issues

- Abbreviations should be used consistently. For example, BC is abbreviated at the start of the Introduction, but then the full form is present on several occasions thereafter. Check other abbreviations.

-          Take care of italicization – in vitro/in vivo/in situ/ex situ (italic). Also italicize the names of microbial species.

-          Take care of capitalization – in the first sentence of the abstract. You do not need to capitalize the first letter of each word. Check it elsewhere.

-          Better avoid abbreviating terms in titles.

Author Response

We would like to thank you for time and energy you invested in our manuscript revision. Your comments and suggestions are much appreciated and we are positive that this manuscript is significantly improved after your remarks consideration and implementation. Pease find attached revised manuscript.

Kind regards,

Marija Mojicevic

Reviewer 2 Report

Overall, the writing of this article has applied reliable scientific principles. The authors using the relevant methods to prove several evidence about the raised topic. However, several confirmation and revision are needed to improve the quality of manuscript.  I put several comments on the revised manuscript.

Author Response

We would like to thank you for time and energy you invested in our manuscript revision. Your comments and suggestions are much appreciated and we are positive that this manuscript is significantly improved after your remarks consideration and implementation. Pease find our responses attached. 

Kind regards,

Marija Mojicevic

Reviewer 3 Report

Manuscript titled " Enhanced Antimicrobial Activity of Biocompatible Bacterial Cellulose Films Via Dual Synergistic Action of Curcumin and Triangular Silver Nanoplates" written by Lanzagorta-Garcia et al. focuses on the biological evaluation of new Synergistic biopolymer as antibacterial agents. The results obtained by the authors are interesting and clearly presented. The in vivo studies represent a relevant added value since the lack of in vivo results in this research field is one of the main prominent problems for the development of the new antibiofilm compound. Nevertheless, some issues need to be fixed to reach the quality required for publication in the Int. J. Mol. Sci. 1. The authors only evaluated two bacteria (Escherichia coli and Staphylococcus aureus). I believe it is not sufficient, and they must test other bacteria. 2. I recommend that antibiofilm activity will be investigated. 3. Bibliography needs an update; some recent 2022 references on the topic should be added: a. Comini, Sara, et al. "Combination of Poly (ε-Caprolactone) Biomaterials and Essential Oils to Achieve Anti-Bacterial and Osteo-Proliferative Properties for 3D-Scaffolds in Regenerative Medicine." Pharmaceutics 14.9 (2022): 1873. b. Halimehjani, Azim Ziyaei, et al. "Synthesis of novel antibacterial and antifungal dithiocarbamate-containing piperazine derivatives via re-engineering multicomponent approach." Heliyon (2022): e09564. c. Wang, Cong, et al. "Cellulose nanofibers aerogels functionalized with AgO: Preparation, characterization and antibacterial activity." International Journal of Biological Macromolecules 194 (2022): 58-65. d. Onyszko, M., et al. "The cellulose fibers functionalized with star-like zinc oxide nanoparticles with boosted antibacterial performance for hygienic products." Scientific Reports 12.1 (2022): 1-13. e. Guo, Chuanpan, et al. "Multimodal Antibacterial Platform Constructed by the Schottky Junction of Curcumin‐Based Bio Metal–Organic Frameworks and Ti3C2Tx MXene Nanosheets for Efficient Wound Healing." Advanced NanoBiomed Research (2022): 2200064. f. Hooshmand, Seyyed Emad, et al. "Antibacterial, antibiofilm, anti-inflammatory, and wound healing effects of nanoscale multifunctional cationic alternating copolymers." Bioorganic Chemistry 119 (2022): 105550.

Author Response

(The authors gave the same response as above.)

Round 2

Reviewer 1 Report

Accept

Reviewer 3 Report

I believe this article could be published in this form; however, the authors must mention in the conclusion part that further investigation like antibiofilm will be tested in the future.